# Determination of the Chloride Ion Deposition by the Bresle Method

**DOI:** 10.3390/ma17235684

**Published:** 2024-11-21

**Authors:** Miroslav Vacek, Vít Křivý, Barbora Křistková

**Affiliations:** Department of Structures, Faculty of Civil Engineering, VSB—Technical University of Ostrava, 708 00 Ostrava, Czech Republic; vit.krivy@vsb.cz (V.K.); barbora.kristkova@vsb.cz (B.K.)

**Keywords:** Bresle method, deposition of chloride ions, wet candle, ISO 9225, ISO 8502-6, ISO 8502-9

## Abstract

In corrosion science, accurate determination of chloride ion deposition rates is critical to mitigating the environmental impact on structures. Traditional methods, such as the wet candle and dry plate methods (ISO 9225), are often inaccurate in capturing localized conditions and are also time-consuming and costly. The Bresle method, which measures soluble salts directly on metal surfaces, offers a more targeted approach. This article examines the Bresle method as an alternative for determining average monthly chloride ion deposition rates, including a regression analysis comparing the Bresle method with the wet candle method, and examines the long-term salinity of exposed surfaces in comparison with the additive approach to surface salinity. This paper hypothesizes that the Bresle method can be used as an alternative to the wet candle method. Linear regression analysis shows a strong correlation in chloride ion deposition rates compared to those measured by the wet candle method. However, cumulative measurements using long-term exposed coupons are unreliable due to inconsistent trends.

## 1. Introduction

The precise quantification of chloride ion deposition rates is of paramount importance in the field of corrosion science. It is also one of aspects of the design and assessment of building structures, with the objective of mitigating the environmental impact on surfaces. This is discussed, for example, in the studies [1,2,3]. Conventional techniques, such as the wet candle and dry plate methods outlined in ISO 9225 [4], involve laboratory investigations that utilize titration or spectrometric analysis to measure chloride ion concentrations, as mentioned in the following studies [1,5,6]. However, these methods often lack the precision needed to capture localized conditions and are also quite time-consuming.

The Bresle method, which was first introduced in 1995 through ISO 8502-6 [7] and ISO 8502-9 [8], is used to measure soluble salts on metal surfaces prior to coating, with the aim of preventing adhesion issues [9]. This method has been adopted by organizations such as the US Navy [10] and ASTM [11], and it remains the primary test method. The method is based on measuring the conductivity of salts in water, whereby each salt exhibits a distinctive conductivity–concentration relationship. The relationship for sodium chloride (NaCl) is published in the Handbook of Chemistry and Physics [12]. The Bresle method is fundamental for determining the cleanliness of a surface prior to painting [13]. Ensuring that the surface is adequately clean before coating helps to prevent corrosion due to contamination, as mentioned in the following studies [14,15]. The Bresle method is primarily used to measure the concentration of soluble salts on metal surfaces, which is of paramount importance before applying protective coatings [9].

The prevention of corrosion is of paramount importance for the maintenance of the structural integrity and durability of structures and components situated in the vicinity of roadways, as is mentioned in [16]. The most commonly used methods to prevent corrosion are barrier coatings, galvanization, cathodic protection, corrosion inhibitors, and maintenance. For further information about corrosion prevention, see [17,18,19]. In order to design effective corrosion protection, it is essential to have a detailed understanding of the environmental conditions in which the structure is located, as underscored by [20,21,22].

Chloride ions are the main corrosion factor in the vicinity of roads as underscored by [23,24,25]. The chloride ions deposited on structure surfaces originate from de-icing salt and brine utilized in winter maintenance for roadway passability [26].

The primary objective of investigating surface salinity is to examine corrosion losses in steel surfaces, as presented in [24,27,28], as well as the deterioration of steel rebars in concrete structures, as presented in [29,30]. It is of great importance to determine the rate of chloride deposition in areas of structural detail, as corrosion damage often occurs on these localized surfaces, potentially compromising the reliability of the structure or its components.

This article examines the use of the Bresle method [7,8] to determine average monthly deposition rates of chloride ions in various locations as an alternative to standard methods. Furthermore, a regression analysis [31] is included, which compares the measurements made by the wet candle and Bresle methods [32]. This article highlights the advantages of the Bresle method for chloride ion deposition rate analysis, including its capacity to capture local effects in situ.

## 2. Methodology

This study investigates the ability to measure the average monthly deposition of chloride ions using stands developed for the wet candle method, in accordance with the ISO 9225 standard [4]. These stands are situated in the vicinity of the I/11 highway between Ostrava and Opava in the Czech Republic, at distances ranging from 3 m to 180 m (see Figure 1). Each stand is marked in Figure 2 for the locality Hrabyně, in Figure 3 for the locality Vřesinská, and in Figure 4 for the locality 17. listopadu. This highlights the fact that the data were collected at different locations and at different distances from the pollution source. This is the authors’ attempt to exclude local effects of individual sites. They are located at 3 independent sites, with a total of 14 stands.

All stands are oriented in accordance with the measured surface area of the coupon relative to the source of chloride ion deposition. A selected typical stand located in the Vřesinská locality situated in the road cut is presented in the Figure 5. For a detailed illustration of the scheme, please see Figure 6. All stands are oriented according to the scheme. This paper presents the findings of measurements conducted on a vertical surface oriented in alignment with the source of deposition, which is situated in a sheltered condition under the roof of stands.

### 2.1. Average Monthly Deposition of Chloride Ions

The average monthly deposition of chloride ions is determined by the wet candle method, as outlined in ISO 9225 [4]. The equation presented in the ISO 9225 [4] standard for determining the average monthly deposition of chloride ions by the wet candle method is
(1)$$\begin{array}{c} \begin{array}{c} S_{d}=\frac{m}{A\cdot t} \end{array} \end{array}$$
where
$S_{d}$—average monthly deposition of chloride ions ($\mathrm{mg}/(m^{2}\cdot \mathrm{day})$);*m*—mass of chloride ions in the solution of 1 L volume (mg);*A*—exposed area ($m^{2}$);*t*—exposure time (days).

### 2.2. Surface Salinity

The surface salinity is determined by the Bresle method in accordance with ISO 8502-6 [7] and ISO 8502-9 [8]. The surface salinity coupons (short-term as well as long-term) are manufactured from carbon steel with zinc-coated protection. Prior to the exposure period and following each measurement, the surface salinity coupon is cleaned with deionized water before the exposure period and after each measurement. The coupon is situated in a sheltered condition to prevent rain from washing the exposed surface.

To ascertain the conductivity of the soluble solution, Defelsko Positector^®^ SST with Defelsko PosiPatch and deionized water are used. The in situ measurement conducted on a short-term coupon is illustrated in Figure 7.

The final equation presented in the ISO 8502-9 [8] standard for the total amount of salt on the surface, as determined by the Bresle method, is as follows:(2)$$\begin{array}{c} \begin{array}{c} \rho_{A}=\frac{m}{A} \end{array} \end{array}$$
where *m* is determined as
(3)$$\begin{array}{c} \begin{array}{c} m=c\cdot V\cdot \Delta \gamma \end{array} \end{array}$$
where Δγ is determined as
(4)$$\begin{array}{c} \begin{array}{c} \Delta \gamma =\gamma_{2}-\gamma_{1} \end{array} \end{array}$$
where
Δγ—change of conductivity (μS/m);$\gamma_{1}$—conductivity of water before test (μS/m);$\gamma_{2}$—conductivity of water after test (μS/m);*V*—volume (mL);*c*—constant for salts on surface—approximately 5 $\mathrm{kg}/({\mathrm{Sm}}^{2})$ 25 °C ($\mathrm{mg}/{\mathrm{Sm}}^{2}$);*m*—mass of extracted salt from the surface in the cell (mg);*A*—area of the cell ($m^{2}$);$\rho_{A}$—total amount of salt on the surface ($\mathrm{mg}/m^{2}$).

The short-term coupon data =set is examined in order to ascertain the total salinity levels that were recorded during the same time period as that of the wet candle (one month). It is not possible to make a direct comparison between the average monthly deposition of chloride ions and the surface salinity levels that were measured on the short-term coupon without first adding the unit of time to the surface salinity value. The following equation is used:(5)$$\begin{array}{c} \begin{array}{c} S_{d,b}=\frac{\rho_{A}}{t} \end{array} \end{array}$$
where
$S_{d,b}$—total amount of salt on the surface recalculated to one day ($\mathrm{mg}/(m^{2}\cdot \mathrm{day})$);$\rho_{A}$—total amount of salt on the surface ($\mathrm{mg}/m^{2}$);*t*—time (days).

### 2.3. Cumulative Measurement

Additionally, the long-term salinity of the surface can be quantified through the implementation of two distinct methodologies. The initial approach entails the utilization of prolonged measurements on a sizable long-term coupon, whereas the alternative involves the aggregation of data obtained from short-term coupons, thereby facilitating an additive analysis. The outcomes of these two techniques are then compared with one another, allowing for a comprehensive evaluation of the long-term measurements in conjunction with the additive sum of measurements derived from the short-term coupon.

#### 2.3.1. Cumulative Long-Term Coupon

In order to obtain a cumulative measurement of surface salinity using the Bresle method, a larger long-term coupon (see Figure 8) is employed, comprising subsurfaces (marked boxes) for each measurement. One of these boxes is measured sequentially over time on the same day that the wet candle is observed.

#### 2.3.2. Additive Sum of Total Amount of Salt on the Surface—Additive Approach

The additive approach adds together the measurements taken by Section 2.2. The additive sum of measurements of surface salinity on the short-term coupon is determined by the following equation:(6)$$\begin{array}{c} \begin{array}{c} S_{d,b,add,cumulative}=\sum_{1^{st}\;month}^{end\;month}S_{d,b}\cdot t \end{array} \end{array}$$


$S_{d,b,add,cumulative}$—additive sum of the total amount of salt on the surface ($\mathrm{mg}/m^{2}$);$S_{d,b}$—total amount of salt on the surface recalculated to one day ($\mathrm{mg}/(m^{2}\cdot \mathrm{day})$);*t*—time (days).


## 3. Results and Discussion

### 3.1. Average Monthly Deposition of Chloride Ions and Surface Salinity

A total of 33 comparable experimental measurements of the average monthly deposition of chloride ions, determined by the wet candle method, and surface salinity, measured by the Bresle method, were analyzed using a linear regression model. The results of this analysis are presented in Figure 9 and also include confidence and prediction intervals.

The comparison of experimental data obtained by the wet candle method and the Bresle method for monthly surface exposure shows a strong correlation (0.8047) with the chloride deposition rate results determined by the traditional wet candle method. Therefore, the Bresle method can be used with sufficient accuracy as an alternative for measuring chloride ion deposition rates, particularly in cases where corrosion processes on surfaces of spatially restricted structural details require analysis. This is of paramount importance, as local corrosion damage in the area of bridge structure details is one of the most common failures that can affect the long-term reliability of the structure or its components. The final equation for the surface salinity and deposition of chloride ions for vertical surfaces is as follows:(7)$$\begin{array}{c} \begin{array}{c} S_{d,b}=23.832\cdot S_{d}-21.192 \end{array} \end{array}$$
where
$S_{d,b}$—total amount of salt on the surface recalculated to one day determined by the Bresle method ($\mathrm{mg}/(m^{2}\cdot \mathrm{day})$);$S_{d}$—average monthly deposition of chloride ions determined by the wet candle method ($\mathrm{mg}/(m^{2}\cdot \mathrm{day})$).

The results of the experimental measurement indicate that it is possible to measure the average monthly deposition of chloride ions on vertical surfaces using the Bresle method. In order to prevent the washing away of the surface by rain, it is necessary to place the coupons in a sheltered condition. The impact of rainfall on the surface salinity is presented in the study [33]. Prior to the exposure period, the coupons should be cleaned with deionized or distilled water.

### 3.2. Cumulative Measurement

The subsequent sections present the findings of the cumulative long-term coupon and additive sum of the total amount of salt on the surface as an additive approach. For the exact position of each stand, see Figure 2, Figure 3 and Figure 4 or the overview map in Figure 1.

#### 3.2.1. Cumulative Long-Term Coupon

The data pertaining to the cumulative coupon are presented in Figure 10. It should be noted that the values may fluctuate in response to weather conditions, particularly in the long term. This phenomenon is discussed in greater detail in Section 3.2.3. Additionally, it is important to note that some measurements are missing due to coupon damage, such as those for month 12/23 in the case of V6-C or month 3/24 in the case of V1-C. The notation “-C” after the locality mark indicates a long-term cumulative coupon.

#### 3.2.2. Additive Sum of Total Amount of Salt on the Surface—Additive Approach

The individual measurements for each month are presented in Figure 11 and the additive cumulative measurements, as outlined in Section 2.3.2, are presented in Figure 12. As Equation (Equation 6) is used, the values are solely increasing. Some measurements are absent (e.g., month 3/24 of the V3-V measurement or months 12/23 and 1/24 of the V5-V measurement) due to damage to the coupon. The notation “-V” after the locality mark indicates a short-term vertical coupon.

It can be seen that there is a considerable impact of the distance between the pollution source and the stand under study. The stands marked V1 to V4 are situated on a road cut in close proximity to the I/11 highway, ranging from 3 to 13 m, and their measured values are high. In contrast, the stand marked H5 is located 180 m from the I/11 highway and its measured values are significantly lower.

It is also important to note that there are $PM_{2.5}$ and $PM_{10}$ particles, which may have a slight impact on the results [34]. Both PM2.5 and PM10 are composed of metals such as Al, Fe, Zn, Cr, Ti, etc., which is underscored in [35,36]. According to the articles [37], the main components of PM2.5 and PM10 are Na+, Ka+, and Cl−. Two of the main components (Na+ and Cl−) are used in the form of salt (NaCl) for the road maintenance. According to the presented results, this influence is small in the case of one-month measurements, according to the good value of the coefficient of determination. The main contribution to the measured conductivity is, therefore, NaCl. The influence of $PM_{10}$ particles, which contain approximately 7 % chloride ions in the Czech Republic, is discussed in [38].

#### 3.2.3. Comparison of Cumulative Long-Term Coupon Approach and Additive Sum of Total Amount of Salt on the Surface—Additive Approach

A comparison of Figure 10 and Figure 12 reveals that cumulative measurements of surface salinity are not feasible. The experimental results corresponding to longer exposure periods than one month demonstrate that with prolonged exposure of the surface, including environmental influences (moisture condensation, rainfall, wind action) begin to apply and significantly influence the change in chloride concentration on the monitored surface. In the case of vertically placed samples, natural surface cleansing of the surface by condensed water is likely to be the most applicable process. It is not possible to use surface salinity measurements from long-term exposed surfaces (i.e., exposure longer than one month) to determine chloride deposition rates. This variability is likely due to moisture precipitation on the surface of the cumulative coupon.

The highest values of the surface salinity (see Figure 11) were observed at every stand in December 2023, and especially in January 2024. This phenomenon can be attributed to fluctuations in temperature and rainfall. When the temperature exceeds 5 °C, road maintenance is not required, as snowfall and frost are not anticipated. The temperature trend during the reporting period is illustrated in Figure 13. In February and March of 2024 there are high temperatures, so the chloride deposition has a decreasing tendency. As the temperature increases, the trend continues to decrease due to the presence of residual chlorides on the road surface as the temperature increases, reaching background chloride deposition typical of summertime. Rainfall also contributes to this trend, but it is more or less constant over the period considered; see Figure 14. Average monthly RH is shown in Figure 15. The average monthly RH values are not significantly different. Additionally, a decline in chloride deposition is evident in the cumulative coupon following the combined effects of washing due to precipitated moisture and cleaning due to wind; see Figure 10.

### 3.3. Advantages of Using the Bresle Method for Determining Chloride Ion Deposition

To accurately assess the local conditions of evaluated structures, it is essential to identify a method that can determine the deposition of chloride ions on surfaces where traditional methods are inapplicable. One of the motivations for evaluating the efficacy of the Bresle method is to identify appropriate techniques for quantifying deposition rates on specific elements of bridge structures where conventional wet candle measurement techniques are inadequate. One such example is the overhanging end of the bottom flange of a box girder, as shown in Figure 16. This is an example of a structural detail for which there is a lack of consensus regarding the impact of chlorides on the service life of the bridge structure [20].

In contrast, the ISO 9225 standardized methods [4] (dry plate and wet candle methods) are not capable of achieving this level of precision. The diameter under investigation is considerably larger, rendering it unfeasible to capture such localized conditions with the same degree of efficiency as the Bresle method. The Bresle approach also has the added benefit of requiring less time and cost per measurement.

Additionally, noncontact methods may be employed to monitor the concentration of chemical elements on surfaces [40,41] or in water [42]. While these techniques offer high precision, they are not as cost-effective, time-efficient, or mobile as the Bresle method set.

In the authors’ opinion, it is appropriate to validate the results of the application of the Bresle method to the measurement of deposition rate using the wet candle method, which is introduced in ISO 9225 [4]. In addition, other chemical and mechanical analyses (e.g., corrosion product thickness, scotch tape test, or XRD or XRF analysis, which are mentioned, for example, in [27,43], should be performed on the surfaces, which do not seem appropriate for validating the results of the Bresle method, although they provide important additional information.

## 4. Conclusions

This article presents a methodology for measuring the average monthly deposition of chloride ions on vertically oriented surfaces, employing the Bresle method. This approach is particularly useful for determining local corrosion aggressivity in areas where standardized methods, such as the wet candle or dry plate, are inapplicable. The Bresle method also has the advantage of being cost-effective, time-efficient, and mobile, which could be beneficial in certain situations.

A comparison of 33 measurements obtained by the wet candle method and the Bresle method revealed that the vertical surface salinity measurement by the Bresle method can be employed as an alternative to the standardized wet candle method. The coefficient of determination, which assumes a value of 0.8047 in the linear regression analysis, illustrates the robust correlation between the aforementioned methods for the determination of the deposition rate of chloride ions. Subsequent research will investigate the potential for applying this approach to horizontal and inclined surfaces, thereby representing another possible real-world scenarios. Additionally, the text indicates that cumulative measurements of surface salinity using long-term exposed coupons are not a viable approach. This is due to the variability in the trend of cumulative measurements, which do not demonstrate a consistent increase. The final equation, which was derived through the process of linear regression, is as follows:(8)$$\begin{array}{c} \begin{array}{c} S_{d,b}=23.832\cdot S_{d}-21.192 \end{array} \end{array}$$
where
$S_{d,b}$—total amount of salt on the surface recalculated to one day determined by the Bresle method ($\mathrm{mg}/(m^{2}\cdot \mathrm{day})$);$S_{d}$—average monthly deposition of chloride ions determined by the wet candle method ($\mathrm{mg}/(m^{2}\cdot \mathrm{day})$).

These results of the measurements can be also useful for comparison with numerical model investigations of the selected locality. The Bresle method allows for the investigation of local deposition rates, which can then be compared with the model. This aspect will be further investigated by the authors.

The Bresle method has the potential to be adapted and applied in a variety of geographic regions and structures. It may be necessary to consider the sheltered condition of the measured surface, as rainwater on an unprotected surface could potentially wash away the surface, resulting in significantly lower measured values after rain or periods of high moisture. The Bresle method is predicated on the assumption that it produces more accurate results when applied directly to the surface of the structure under examination. This approach enables the detection of localized corrosion phenomena resulting from chloride ions. The Bresle method represents an alternative approach to the measurement of chloride deposition rates in construction details that may be employed in instances where the traditional wet candle method is not a viable option. Furthermore, the Bresle method offers the additional advantage of requiring less time and a lower financial investment per measurement.

One advantage of the Bresle method is that it can be used directly on the surface in the case of steel surfaces. This approach could potentially enhance the understanding of chloride ion deposition on sheltered structures. However, it should be noted that the method is not applicable to nonsheltered surfaces, which could be considered a limitation.

## Figures and Tables

**Figure 1 materials-17-05684-f001:**
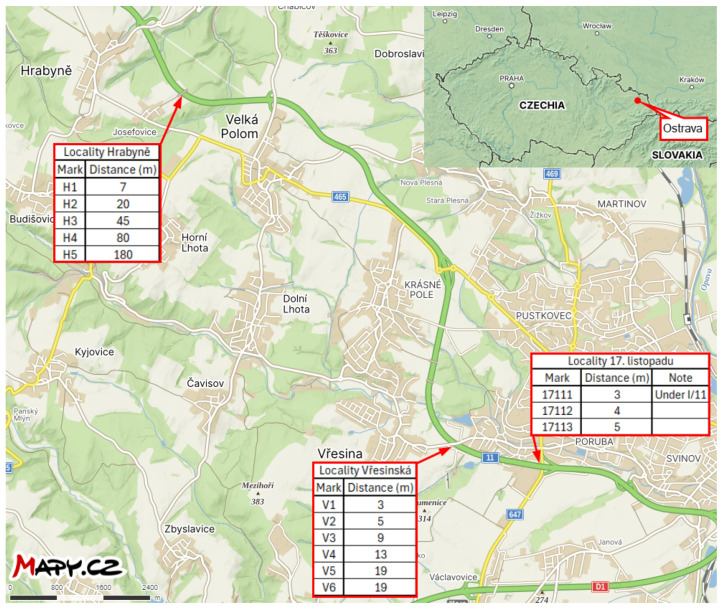
Overview map of the measured localities, Ostrava region, CZ. Source: mapy.cz.

**Figure 2 materials-17-05684-f002:**
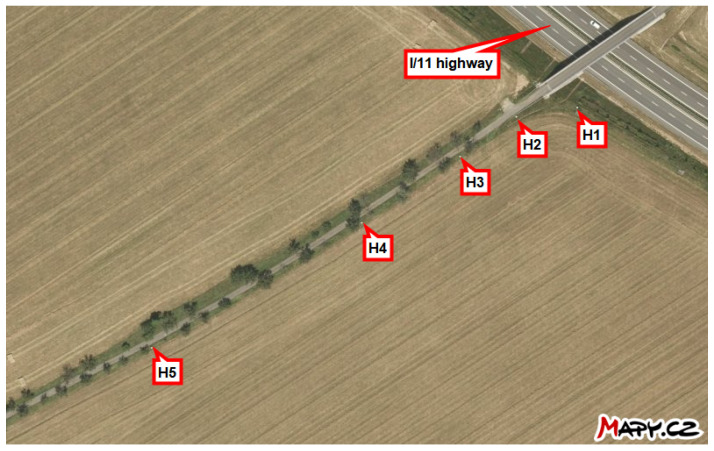
Locality Hrabyně, locality exploring more distant surroundings. Source: mapy.cz.

**Figure 3 materials-17-05684-f003:**
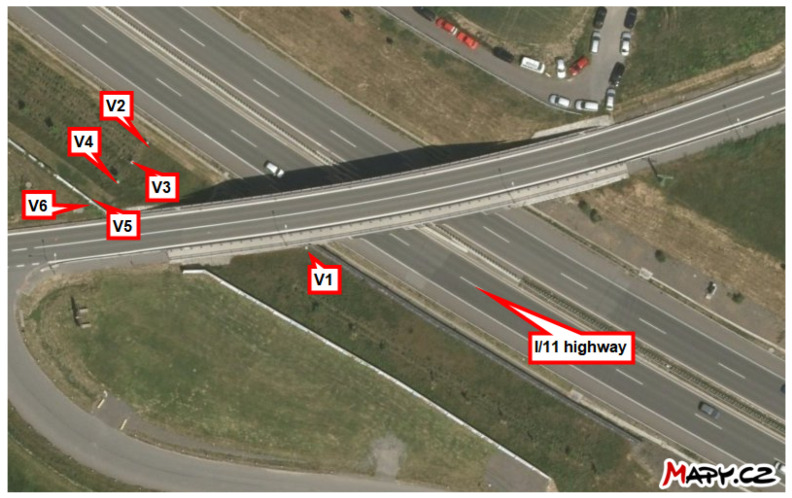
Locality Vřesinská, locality investigating the effect of the slope of the road cut. Source: mapy.cz.

**Figure 4 materials-17-05684-f004:**
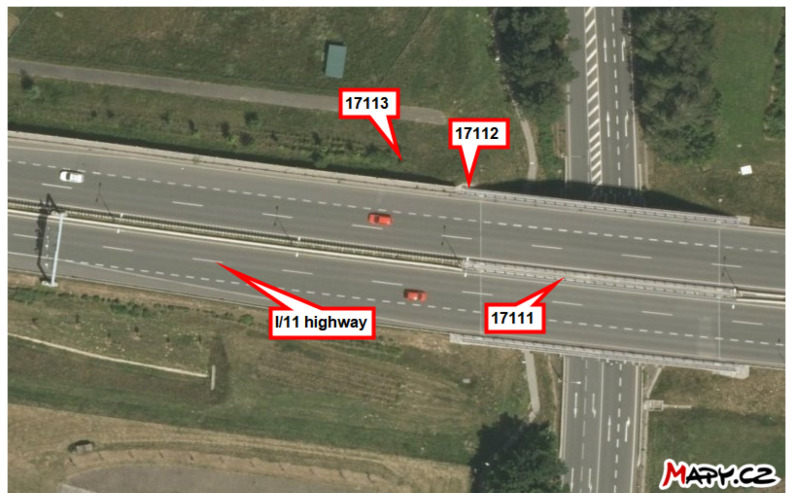
Locality 17. listopadu, locality investigating the very close surroundings, the effect of the noise wall, and the situation under the bridge structure. Source: mapy.cz.

**Figure 5 materials-17-05684-f005:**
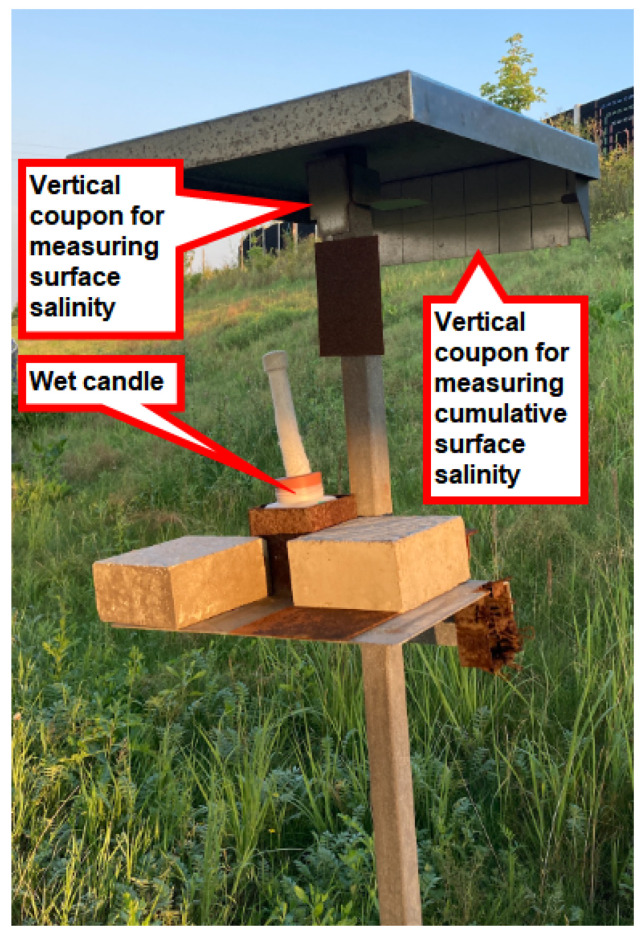
Selected typical stand located in the Vřesinská locality on the slope of the road cut, Ostrava region, CZ.

**Figure 6 materials-17-05684-f006:**
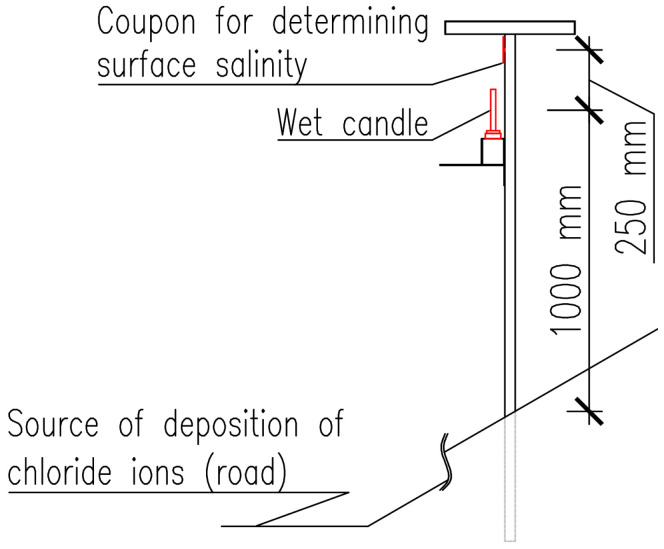
The typical configuration of a stand.

**Figure 7 materials-17-05684-f007:**
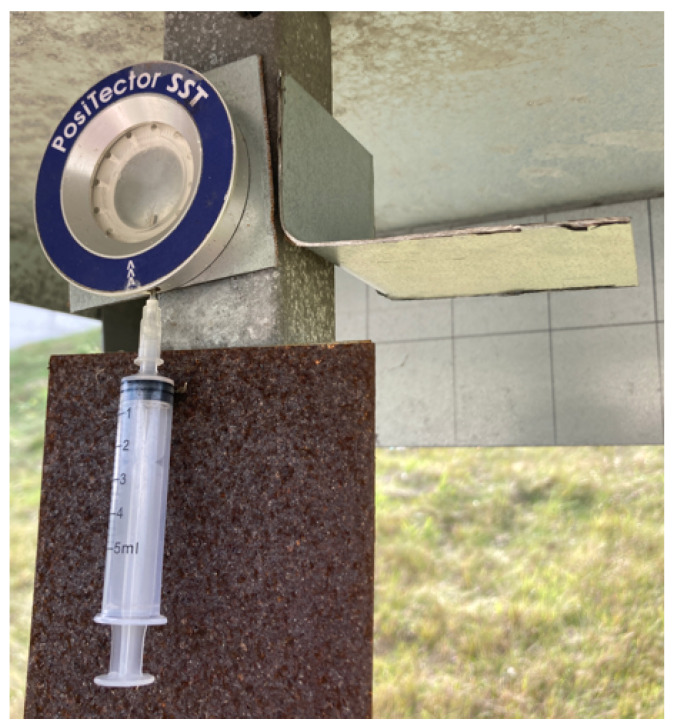
In situ measurement performed on a short-term coupon.

**Figure 8 materials-17-05684-f008:**
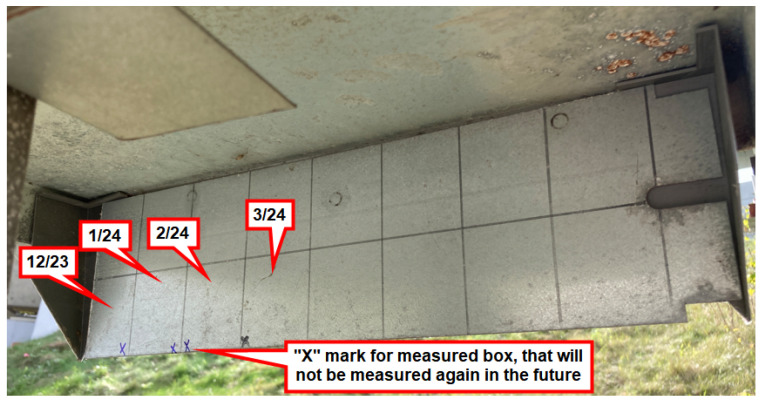
Long-term coupon with subsurfaces (marked boxes), example of a measurement process.

**Figure 9 materials-17-05684-f009:**
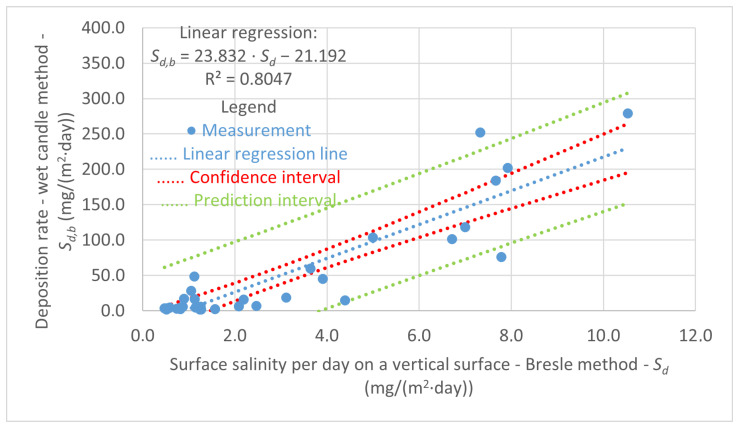
Linear regression of the average monthly deposition of chloride ions measured by the wet candle method and the surface salinity of vertical coupon measured by the Bresle method, 5 % and 95 % confidence interval and prediction interval.

**Figure 10 materials-17-05684-f010:**
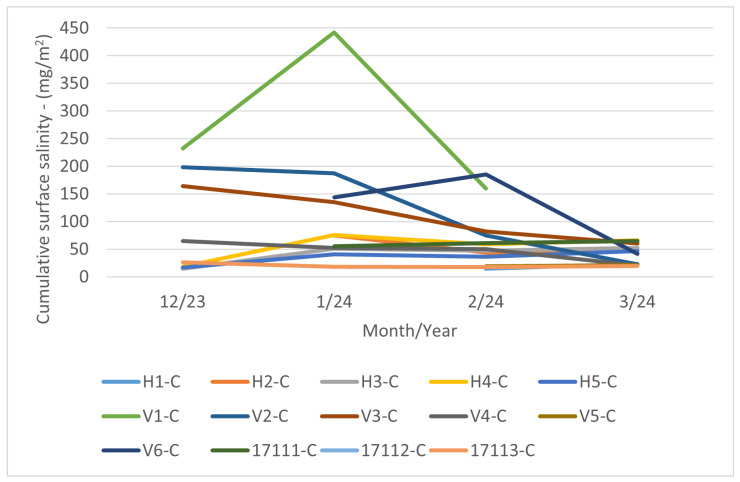
Surface salinity—cumulative vertical coupon.

**Figure 11 materials-17-05684-f011:**
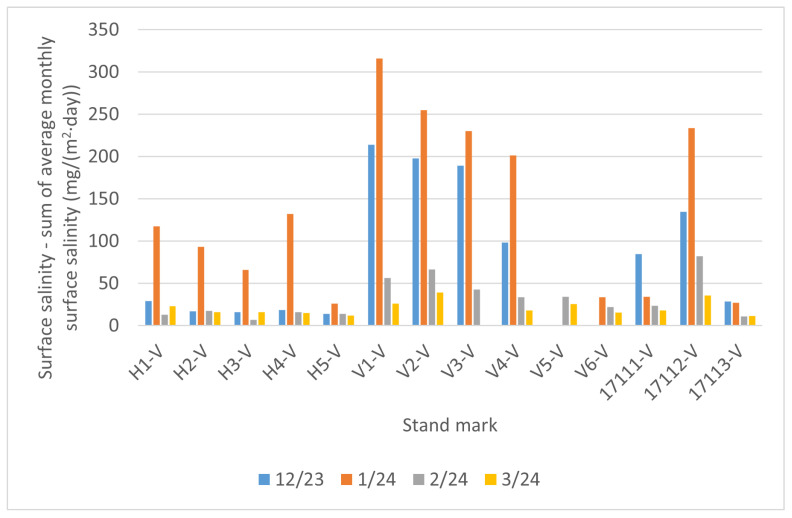
Surface salinity—short-term measurements on vertical coupon by each month.

**Figure 12 materials-17-05684-f012:**
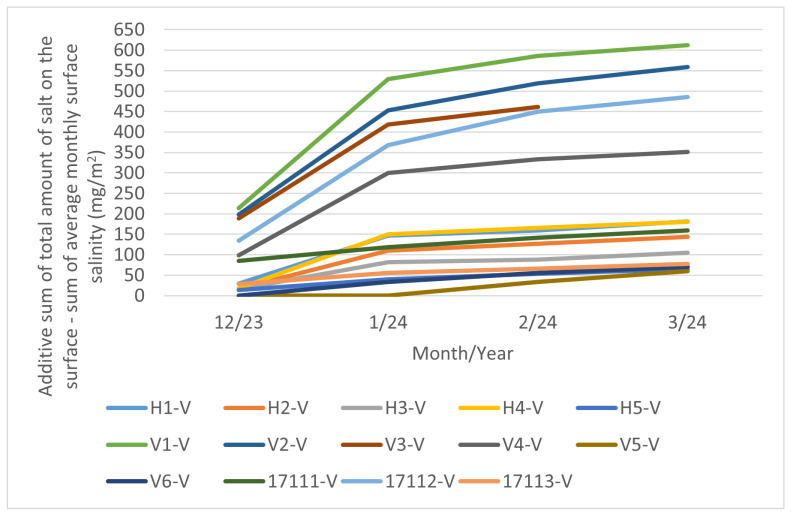
Surface salinity—additive sum of short-term measurements on vertical coupon.

**Figure 13 materials-17-05684-f013:**
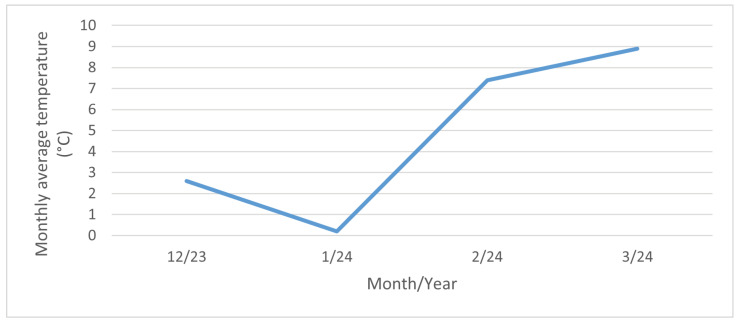
Monthly average temperature. Dataset source: [39].

**Figure 14 materials-17-05684-f014:**
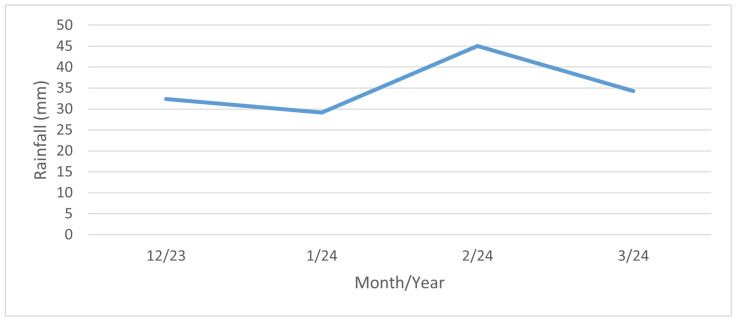
Monthly rainfall. Dataset source: [39].

**Figure 15 materials-17-05684-f015:**
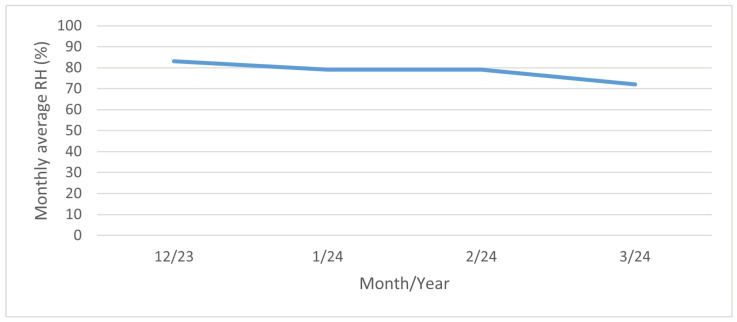
Monthly average RH. Dataset source: [39].

**Figure 16 materials-17-05684-f016:**
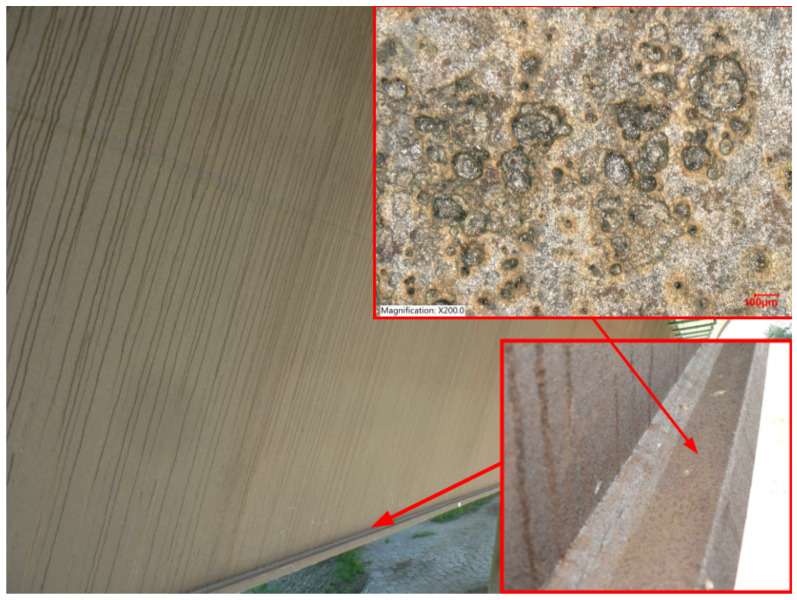
Bridge across the Ostravice river, Czech Republic; photo and detail of the connection of the exterior web plate and bottom slab plate of the box girder; the top surface of the bottom slab of the bridge—magnification 200×, Keyence VHX-7000.

## Data Availability

The data presented in study are available on Zenodo, https://zenodo.org/records/13938046, accessed on 18 October 2024.

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
