# Peer review of "Determination of the Chloride Ion Deposition by the Bresle Method"

_materials, 2024, doi:10.3390/ma17235684_

Round 1
Reviewer 1 Report
Comments and Suggestions for Authors
The paper examines the effectiveness of the Bresle method for measuring chloride ion deposition on metallic surfaces, comparing it with traditional methods such as the wet candle method. The work addresses a significant issue in corrosion science, emphasizing the need for precise methods to evaluate chloride deposition in localized high-risk areas, such as structures near roadways. The methodology and analysis provide a promising alternative to enhance field assessments, although certain methodological and analytical aspects could be refined to strengthen the study's applicability and accuracy.
Test reiterations: It is essential to specify the number of reiterations for each monthly measurement and apply a suitable statistical analysis. This should include the calculation of standard deviations and variability analyses, improving the consistency and accuracy of the results.
Control of environmental factors: thoroughly integrating and documenting environmental conditions, including relative humidity during each measurement, would be beneficial, as these factors significantly impact chloride deposition and the comparability of the data.
Homogeneity in figure sizes: the figure sizes should follow a more consistent criterion; mixing very large figures with relatively small ones gives an impression of low scientific rigor.
Presentation of data in Ffgures 13 and 14: the way data is presented in Figures 13 and 14 is confusing and very difficult to follow. It is recommended that the authors approach this differently.
General applicability of the method: It would be helpful to discuss how the Bresle method could be adapted and applied in different geographic regions and types of structures, allowing for an evaluation of its relevance in a broader, global context within the field of corrosion science.
Author Response
Comments 1: The paper examines the effectiveness of the Bresle method for measuring chloride ion deposition on metallic surfaces, comparing it with traditional methods such as the wet candle method. The work addresses a significant issue in corrosion science, emphasizing the need for precise methods to evaluate chloride deposition in localized high-risk areas, such as structures near roadways. The methodology and analysis provide a promising alternative to enhance field assessments, although certain methodological and analytical aspects could be refined to strengthen the study's applicability and accuracy.
Test reiterations: It is essential to specify the number of reiterations for each monthly measurement and apply a suitable statistical analysis. This should include the calculation of standard deviations and variability analyses, improving the consistency and accuracy of the results.
Response 1: The authors thank the reviewer for his insightful comment. The consistency of measurements by the Bresle method will be the subject of future research. For now, only one comparable measurement is available for each measurement made by the Bresle method and the wet candle method. Confidence and prediction intervals are incorporated into the results.
Comments 2: Control of environmental factors: thoroughly integrating and documenting environmental conditions, including relative humidity during each measurement, would be beneficial, as these factors significantly impact chloride deposition and the comparability of the data.
Response 2: The authors thank the reviewer for his insightful comment. Information about the average monthly RH value was added.
Comments 3: Homogeneity in figure sizes: the figure sizes should follow a more consistent criterion; mixing very large figures with relatively small ones gives an impression of low scientific rigor.
Response 3: The authors thank the reviewer for his insightful comment. Figures are now only 0.8 times the width for maps and horizontally oriented images, 0.5 times the width for vertically oriented images, and 1.0 times the width for graphs.
Comments 4: Presentation of data in Ffgures 13 and 14: the way data is presented in Figures 13 and 14 is confusing and very difficult to follow. It is recommended that the authors approach this differently.
Response 4: The authors thank the reviewer for his insightful comment. Figures 13 and 14 (in the revised version – Figures 11 and 13) illustrate different methodological approaches according to sections 2.3.1 and 2.3.2 respectively. To enhance clarity, the text has been divided into sections 3.2, 3.2.1, and 3.2.2. The names of the different measurement approaches are also highlighted in the text.
Comments 5: General applicability of the method: It would be helpful to discuss how the Bresle method could be adapted and applied in different geographic regions and types of structures, allowing for an evaluation of its relevance in a broader, global context within the field of corrosion science.
Response 5: The authors thank the reviewer for his insightful comments. There is added paragraph about applicability of the Bresle method.
The Bresle method has the potential to be adapted and applied in a variety of geographic regions and structures. It may be necessary to consider the sheltered condition of the measured surface, as rainwater on an unprotected surface could potentially wash away the surface and result in significantly lower measured values after rain or periods of high moisture. The Bresle method is based on the premise that it yields more precise outcomes when applied directly to the surface of the structure under examination. This approach allows for the detection of localized corrosion phenomena resulting from chloride ions. The Bresle method represents an alternative approach to the measurement of chloride deposition rates in construction details that may be employed in instances where the traditional wet candle method is not a viable option.
Reviewer 2 Report
Comments and Suggestions for Authors
The manuscript ID materials-3295468 mainly presents a study about the Bresle method as an alternative for measuring particular chloride ion deposition rates. The authors employed a regression analysis comparing it with the wet candle method. Please see below a list of comments to the authors:
1. It is not clear the importance to present figures 4-7.
2. The selection of the area described in figure 8 should be justified.
3. The influence of temperature over the main findings should be discussed.
4. Please comment about the potential influence of different impurities in measurements reported.
5. Advantages and disadvantages of the method proposed should be summarized in the discussion section.
6. The authors are invited to compare non-contact methods based in sensitive advanced techniques. You can see for instance: https://doi.org/10.1088/1361-6501/ad817d
7. Perspectives could be described with better details. You can see for instance: https://doi.org/10.5006/3622
8. The effects responsible for the non-monotonic behavior in data plotted in figure 13 should be described.
9. The error bar in experimental data must be described.
10. The citations presented in collective form should be split in order to justify the importance of each reference selected for the analysis and presentation of this topic.
Author Response
Comments 1: It is not clear the importance to present figures 4-7.
Response 1: The authors thank the reviewer for his insightful comments. It is added following text in the article:
This highlights the fact that the data were collected at different locations and at different distances from the pollution source. This is the authors' attempt to exclude local effects of individual sites.
Comments 2: The selection of the area described in figure 8 should be justified.
Response 2: The authors thank the reviewer for his insightful comments. The stand shown is a typical stand structure. The same structures can be found on the other stands. Information about that the stand is typical is added.
Comments 3: The influence of temperature over the main findings should be discussed.
Response 3: The authors thank the reviewer for his insightful comments. Influence of temperature is discussed in the text.
When the temperature exceeds 5 °C, road maintenance is not required, as snowfall and frost are not anticipated. The temperature trend during the reporting period is illustrated in the Figure 14. In February and March of 2024 were high temperatures, so the chloride deposition has a decreasing tendency. As the temperature raises, the trend continues to decrease due to the presence of residual chlorides on the road surface as the temperature increases, reaching background chloride deposition typical of summertime.
Comments 4: Please comment about the potential influence of different impurities in measurements reported.
Response 4: The authors thank the reviewer for his insightful comments. There is added paragraph about influence of PM2.5 and PM10.
It is also important to note that there are PM2.5 and PM10 particles, which may have a slight impact on the results [31]. The influence of PM10 particles, which contain approximately 7 % chloride ions in the Czech Republic, is discussed in reference [32].
Comments 5: Advantages and disadvantages of the method proposed should be summarized in the discussion section.
Response 5: The authors thank the reviewer for his insightful comments. There is added paragraph about advantages and disadvantages.
One advantage of the Bresle method is that it can be used directly on the surface in the case of steel surfaces. This could potentially provide a better understanding of the deposition of chloride ions on the sheltered structure. However, it should be noted that the method is not usable on non-sheltered surfaces, which could be seen as a disadvantage.
Comments 6: The authors are invited to compare non-contact methods based in sensitive advanced techniques. You can see for instance: https://doi.org/10.1088/1361-6501/ad817d
Response 6: The authors thank the reviewer for his insightful comments. This will be part of the future research according to the results of long-term exposed coupons.
Comments 7: Perspectives could be described with better details. You can see for instance: https://doi.org/10.5006/3622
Response 7: The authors thank the reviewer for his insightful comments. There is added paragraph about applicability of the Bresle method.
The Bresle method has the potential to be adapted and applied in a variety of geographic regions and structures. It may be necessary to consider the sheltered condition of the measured surface, as rainwater on an unprotected surface could potentially wash away the surface and result in significantly lower measured values after rain or periods of high moisture. The Bresle method is based on the premise that it yields more precise outcomes when applied directly to the surface of the structure under examination. This approach allows for the detection of localized corrosion phenomena resulting from chloride ions. Furthermore, the Bresle method offers the additional advantage of requiring less time and a lower financial investment per measurement.
Comments 8: The effects responsible for the non-monotonic behavior in data plotted in figure 13 should be described.
Response 8: The authors thank the reviewer for his insightful comments. Influence of non-monotonic behaviour is discussed in the text where is discussed .
A comparison of Figures 11 and 13 reveals that cumulative measurements of surface salinity are not feasible. The experimental results corresponding to longer exposure period than one month demonstrate that with prolonged exposure of the surface, including environmental influences (moisture condensation, rainfall, wind action) begin to apply and significantly influence on the change in chloride concentration on the monitored surface. In the case of vertically placed samples, natural surface cleansing of the surface by condensed water is likely to be the most applicable process. It is not possible to use surface salinity measurements from long-term exposed surfaces (i.e., exposure longer than one month) to determine chloride deposition rates. This variability is likely due to moisture precipitation on the surface of the cumulative coupon.
Comments 9: The error bar in experimental data must be described.
Response 9: The authors thank the reviewer for his insightful comments. In the text is not used error bar.
Comments 10: The citations presented in collective form should be split in order to justify the importance of each reference selected for the analysis and presentation of this topic.
Response 10: The authors thank the reviewer for his insightful comments. Collective references are corrected.
Reviewer 3 Report
Comments and Suggestions for Authors
Highlight changes in yellow in a next revision, please. No track changes.
As immediately seen, t English needs correction....
“Determining of the deposition”
Do not refer to the article as if it were a person at least say this study underscores...
Being a scientific article, we need to have a clear context, a clear methodology, clear findings, clear implications. as written, the abstract is rather general and is almost seems a summary from a chapter...
Sing the same as presented in the abstract. Try not to duplicate information through the text. Every information needs to be new.
“1. Introduction In the field of corrosion science and subsequent applications in the design and assess- ment of building structures, accurately determining the deposition rate of chloride ions is of great importance for understanding and mitigating the effects of environmental factors on structure surfaces”
Make captions as enlightening as possible. For example, add the time frame.
“Figure 1. Bridge across the Ostravice river, Czech Republic; photo and detail of the connection of the exterior web plate and bottom slab plate of the box girder”
I am not sure I understand why are these figures being presented in an introductory section...
“Figure 2. Corrosion coupon situated at the top surface of the bottom slab of the bridge after one year exposure time - pitting corrosion - magnification 200x, Keyence VHX-7000”
That is, if belonging to the authors, they could integrate discussion, perhaps, not here.
Captions more self explanatory for example add the region.
“Figure 4. Map of the measured localities, source: mapy.cz”
There is little interest in seeing figures where the caption almost seems alike. In that case, you should group them...
“Figure 6. Map of locality VÅ™esinská, source: mapy.cz”
etc
?!
Focusing on the caption alone, the reader will not understand.
“Figure 8. Selected stand located in the VÅ™esinská locality”
“Figure 9. Scheme of typical stand”
Equations presented without an immediate reference before means complete originality, assure that...
“The equation for determining the average monthly deposition of chloride ions by the wet candle method is: Sd = A·t m (1) where • Sd - average monthly deposition of chloride ions (mg/(m2 · day)), • m - mass of chloride ions in the solution of 1 liter volume (mg), • A - exposed area (m2), • t - exposure time (days).”
check them all
Revised use of the italics inside the figure. Also replace the * by the proper symbol in multiplication...
“Figure 12. Linear regression of the average monthly deposition of chloride ions measured by the wet candle method and the surface salinity of vertical coupon measured by the Bresle method”
There should be no mention to another figure In a figures caption, also. as presented, the results will not allow to understand what are the experimental values.
There is also difficulty in linking the legend as authors should find a way to define it again in the caption itself, So not to force the reader to look for it.
“Figure 13. Surface salinity - cumulative vertical coupon; for stand marking, see Figure 4”
I do not really understand in this case why separate the discussion from the results. The discussion needs to rely on references to support or oppose what is being discussed then. There is also a problem with the new graphics being presented in the discussion. This is the place to discuss not to present data...
I urge the authors to merge the discussion with the results because of the involve statements relying on references.
Please check international unit system everywhere:
“H5 is located 180 meters”
Unfortunately, and once again, the conclusions section is written in the same style used in the abstract. So again, we need to have context, methods, findings, implications and in this particular case, limitations and future prospects.
If authors are presenting quantitative data, that should be translated in both the abstract as in the conclusions section, which together need to translate the relevance of the paper.
Without these corrections, it is difficult to access the coherence of the entire paper and the work being presented.
References are very scarce, they need updating: 2024... and more international references, more articles too...
Comments on the Quality of English Language
revise English and language
Author Response
Comments 1: As immediately seen, t English needs correction....
“Determining of the deposition”
Response 1: The authors thank the reviewer for his insightful comments. English language correction was carried out.
Comments 2: Do not refer to the article as if it were a person at least say this study underscores...
Response 2: The authors thank the reviewer for his insightful comments. References are corrected.
Comments 3: Being a scientific article, we need to have a clear context, a clear methodology, clear findings, clear implications. as written, the abstract is rather general and is almost seems a summary from a chapter...
Response 3: The authors thank the reviewer for his insightful comments. Abstract is improved.
In corrosion science, accurate determination of chloride ion deposition rates is critical to mitigating the environmental impact on structures. Traditional methods, such as the wet candle and dry plate methods (ISO 9225), are often inaccurate in capturing localized conditions and are also time consuming and costly. The Bresle method, which measures soluble salts directly on metal surfaces, offers a more targeted approach. This article examines the Bresle method as an alternative for determining average monthly chloride ion deposition rates, including a regression analysis comparing the Bresle method with the wet candle method, and examines the long-term salinity of exposed surfaces in comparison with the additive approach to surface salinity. The paper hypothesizes that the Bresle method can be used as an alternative to the wet candle method. Linear regression analysis shows a strong correlation in chloride ion deposition rates compared to those measured by the wet candle method. However, cumulative measurements using long-term exposed coupons are unreliable due to inconsistent trends.
Comments 4: Sing the same as presented in the abstract. Try not to duplicate information through the text. Every information needs to be new.
“1. Introduction In the field of corrosion science and subsequent applications in the design and assess- ment of building structures, accurately determining the deposition rate of chloride ions is of great importance for understanding and mitigating the effects of environmental factors on structure surfaces”
Response 4: The authors thank the reviewer for his insightful comments. Introduction is improved.
The accurate determination of chloride ion deposition rates is of paramount importance in the field of corrosion science, as well as being a fundamental aspect of the design and assessment of building structures, with the objective of mitigating the environmental impact on surfaces.
Comment 5: Make captions as enlightening as possible. For example, add the time frame.
“Figure 1. Bridge across the Ostravice river, Czech Republic; photo and detail of the connection of the exterior web plate and bottom slab plate of the box girder”
Response 5: The authors thank the reviewer for his insightful comments. This figure outlines a local problem where it would be beneficial to study the deposition of chloride ions in situ. However, it is possible that pure standard methods may not be entirely appropriate for this situation. This figure is not intended to be instructive, but to indicate a possible common situation that should be investigated. One of the reasons for testing the use of the Bresl method is to find suitable procedures for measuring deposition rates on bridge structural details where traditional wet candle measurement sets cannot be applied with enough accuracy. Figure 1 has been included in the paper as a selected example of such a detail (the overlap of the bottom flange of a chamber bridge) for which there is some debate about the effect of chlorides on the service life of the bridge structure.
The paragraph preceding the Figure was rewritten as follows:
To accurately assess the local conditions of evaluated structures, it is essential to identify a method that can determine the deposition of chloride ions on surfaces where traditional methods are inapplicable. One of the motivations for evaluating the efficacy of the Bresle method is to identify appropriate techniques for quantifying deposition rates on specific elements of bridge structures where conventional wet candle measurement techniques are inadequate. One such example is the overhanging end of the bottom flange of a box girder, as shown in Figure 1. This is an example of a structural detail for which there is a lack of consensus regarding the impact of chlorides on the service life of the bridge structure [24].
Comments 6: I am not sure I understand why are these figures being presented in an introductory section...
“Figure 2. Corrosion coupon situated at the top surface of the bottom slab of the bridge after one year exposure time - pitting corrosion - magnification 200x, Keyence VHX-7000”
That is, if belonging to the authors, they could integrate discussion, perhaps, not here.
Response 6: The authors thank the reviewer for his insightful comments. The figures have been reduced solely to Figure 1, which, following an edit, provides a more comprehensive and readily understandable illustration of the surface in question.
Captions are more self-explanatory, such as adding the region.
Comments 7: “Figure 4. Map of the measured localities, source: mapy.cz”
There is little interest in seeing figures where the caption almost seems alike. In that case, you should group them...
“Figure 6. Map of locality VÅ™esinská, source: mapy.cz”
etc.?!
Focusing on the caption alone, the reader will not understand.
“Figure 8. Selected stand located in the VÅ™esinská locality”
“Figure 9. Scheme of typical stand”
Response 7: The authors thank the reviewer for his insightful comments. Captions of the Figures was improwed as follows:
Figure 2. Map of the measured localities, Ostrava region, CZ, source: mapy.cz
Figure 3. Map of locality Hrabynˇe, locality exploring more distant surroundings, source: mapy.cz
Figure 4. Map of locality Vˇresinská, locality investigating the effect of the slope of the road cut, source: mapy.cz
Figure 5. Map of locality 17. listopadu, locality investigating the very close surroundings, the effect of the noise wall and the situation under the bridge structure, source: mapy.cz
In addition, the following has been added to the previous paragraph:
This highlights the fact that the data were collected at different locations and at different distances from the pollution source. This is the authors' attempt to exclude local effects of individual sites.
Figure 6. Selected typical stand located in the VÅ™esinská locality on the slope of the road cut, Ostrava region, CZ
Figure 7. The typical configuration of a stand
Furthermore, the preceding paragraph, which pertains to the figures, has been enhanced. The figures in question depict exposed coupons and the standardized methodology for measuring the rate of chloride ion deposition.
All stands are oriented in accordance with the measured surface area of the coupon relative to the source of chloride ion deposition. Selected typical stand located in the VÅ™esinská locality situated in the road cut is presented in the Figure 6. For a detailed illustration of the scheme, please direct your attention to Figure 7. All stands are oriented according to the scheme. This paper presents the findings of measurements conducted on a vertical surface oriented in alignment with the source of deposition, which is situated in sheltered condition under the roof of stands.
Comments 8: Equations presented without an immediate reference before means complete originality, assure that...
“The equation for determining the average monthly deposition of chloride ions by the wet candle method is: Sd = A·t m (1) where • Sd - average monthly deposition of chloride ions (mg/(m2 · day)), • m - mass of chloride ions in the solution of 1 liter volume (mg), • A - exposed area (m2), • t - exposure time (days).”
check them all
Response 8: The authors thank the reviewer for his insightful comments. Presented equations are revised and there are added references directly before the equations.
Comments 9: Revised use of the italics inside the figure. Also replace the * by the proper symbol in multiplication...
“Figure 12. Linear regression of the average monthly deposition of chloride ions measured by the wet candle method and the surface salinity of vertical coupon measured by the Bresle method”
Response 9: The authors thank the reviewer for his insightful comments. Italic inside figures and usage of * are revised.
Comments 10: There should be no mention to another figure In a figures caption, also. as presented, the results will not allow to understand what are the experimental values.
There is also difficulty in linking the legend as authors should find a way to define it again in the caption itself, So not to force the reader to look for it.
“Figure 13. Surface salinity - cumulative vertical coupon; for stand marking, see Figure 4”
Response 10: The authors thank the reviewer for his insightful comments. The reference to another figure is deleted and replaced by the following text in the preceding text:
The subsequent sections present the findings of the cumulative long-term coupon and additive sum of the total amount of salt on the surface as an additive approach. For the exact position of each stand see Figures 3, 4 and 5 or overview map in the Figure 2.
Comments 11: I do not really understand in this case why separate the discussion from the results. The discussion needs to rely on references to support or oppose what is being discussed then. There is also a problem with the new graphics being presented in the discussion. This is the place to discuss not to present data...
I urge the authors to merge the discussion with the results because of the involve statements relying on references.
Response 11: The authors thank the reviewer for his insightful comments. Results and Discussion are merged. The relevant paragraphs from the Results and Discussion sections are included in the edited version of the article in logical continuity with the text.
Comments 12: Please check international unit system everywhere:
“H5 is located 180 meters”
Response 12: The authors thank the reviewer for his insightful comments. Meters are replaced by only m.
Comments 13: Unfortunately, and once again, the conclusions section is written in the same style used in the abstract. So again, we need to have context, methods, findings, implications and in this particular case, limitations and future prospects.
Response 13: The authors thank the reviewer for his insightful comments. Conclusion section is improved.
Comments 14: If authors are presenting quantitative data, that should be translated in both the abstract as in the conclusions section, which together need to translate the relevance of the paper.
Without these corrections, it is difficult to access the coherence of the entire paper and the work being presented.
Response 14: The authors thank the reviewer for his insightful comments. Both abstract and conclusion sections are improved.
Comments 15: References are very scarce, they need updating: 2024... and more international references, more articles too...
Response 15: The authors thank the reviewer for his insightful comments. References are revised and added a few actual publications related to the topic of the article.
Comments 16: Comments on the Quality of English Language
revise English and language
Response 16: The authors thank the reviewer for his insightful comments. English language correction was carried out.
Round 2
Reviewer 2 Report
Comments and Suggestions for Authors
I appreciate the effort of the authors to improve the presentation of their work. However, fundamental issues are still present. Please see below
*The potential influence of different impurities in measurements reported should be better supported.
*Advantages and disadvantages of the method proposed should be compared with non-contact methods based in sensitive advanced techniques in order to see the value of the work.
Reviewer 3 Report
Comments and Suggestions for Authors
Highlight changes in yellow in a next revision, please. No track changes.
I must say, I do not even understand these kind of answers.
In a next time, please add more contextualizing answers to the clear comments being made.
“Response 2: The authors thank the reviewer for his insightful comments. References are corrected.”
How?! I see NO changes whatsoever in yellow, nor quantitative results:
“Response 3: The authors thank the reviewer for his insightful comments. Abstract is improved.”
“Abstract: In corrosion science, accurate determination of chloride ion deposition rates is critical to mitigating the environmental impact on structures. Traditional methods, such as the wet candle and dry plate methods (ISO 9225), are often inaccurate in capturing localized conditions and are also time consuming and costly. The Bresle method, which measures soluble salts directly on metal surfaces, offers a more targeted approach. This article examines the Bresle method as an alternative for determining average monthly chloride ion deposition rates, including a regression analysis comparing the Bresle method with the wet candle method, and examines the long-term salinity of exposed surfaces in comparison with the additive approach to surface salinity. The paper hypothesizes that the Bresle method can be used as an alternative to the wet candle method. Linear regression analysis shows a strong correlation in chloride ion deposition rates compared to those measured by the wet candle method. However, cumulative measurements using long-term exposed coupons are unreliable due to inconsistent trends.”
So general, where are the references then?
“The accurate determination of chloride ion deposition rates is of paramount importance in the field of corrosion science, as well as being a fundamental aspect of the design and assessment of building structures, with the objective of mitigating the environmental impact on surfaces.”
Again, I am sorry and I must insist. But if this is an introductory section, either the figure is original or not. If it’s not, we need to have references. If it is original, it makes no sense to be in the introductory section.
“Figure 1. Bridge across the Ostravice river, Czech Republic; photo and detail of the connection of the exterior web plate and bottom slab plate of the box girder; the top surface of the bottom slab of the bridge - magnification 200x, Keyence VHX-7000”
Please do not include the term map at the beginning of the caption.
“Figure 3. Map of lo”
etc
Why include equations here that are not original and are not absolutely necessary to be present?
Check all other cases.
“The equation presented in the ISO 9225 [1] standard for determining the average monthly deposition of chloride ions by the wet candle method is:”
I am sorry, but we clearly need to have an occasion per line and present it next to the line. Also address italics It is all the way other way around, for example, units are not to be in italics.
“Figure 10. Linear regression of the average monthly deposition of chloride ions measured by the wet candle method and the surface salinity of vertical coupon measured by the Bresle method, 5 % and 95 % confidence interval and prediction interval”
After merging the discussion with the results, references are very scarcely used, and I do not, for example, understand the presence of the reference in the caption of figure 16 which is not justified, that is adapted data collected from, what?
“Figure 16. Monthly average RH [33]”
Conclusions need to start with a context then we need to have quantitative data because of the nature of this paper.
The number of references are very scarred. Many of them are rather old. You should improve them and add much more recent, relevant and international references.
Comments on the Quality of English Language
moderate
